# Corneal Surface Wave Propagation Associated with Intraocular Pressures: OCT Elastography Assessment in a Simplified Eye Model

**DOI:** 10.3390/bioengineering10070754

**Published:** 2023-06-24

**Authors:** Guoqin Ma, Jing Cai, Rijian Zhong, Weichao He, Haoxi Ye, Chaitanya Duvvuri, Chengjin Song, Jinping Feng, Lin An, Jia Qin, Yanping Huang, Jingjiang Xu, Michael D. Twa, Gongpu Lan

**Affiliations:** 1School of Mechatronic Engineering and Automation, Foshan University, Foshan 528000, China; 2Guangdong-Hong Kong-Macao Intelligent Micro-Nano Optoelectronic Technology Joint Laboratory, School of Physics and Optoelectronic Engineering, Foshan University, Foshan 528000, China; 3College of Optometry, University of Houston, Houston, TX 77204, USA; 4Institute of Engineering and Technology, Hubei University of Science and Technology, Xianning 437100, China; 5Weiren Meditech Co., Ltd., Foshan 528000, China

**Keywords:** corneal biomechanics, optical coherence elastography, optical coherence tomography, intraocular pressure, mechanical wave propagation

## Abstract

Assessing corneal biomechanics in vivo has long been a challenge in the field of ophthalmology. Despite recent advances in optical coherence tomography (OCT)-based elastography (OCE) methods, controversy remains regarding the effect of intraocular pressure (IOP) on mechanical wave propagation speed in the cornea. This could be attributed to the complexity of corneal biomechanics and the difficulties associated with conducting in vivo corneal shear-wave OCE measurements. We constructed a simplified artificial eye model with a silicone cornea and controllable IOPs and performed surface wave OCE measurements in radial directions (54–324°) of the silicone cornea at different IOP levels (10–40 mmHg). The results demonstrated increases in wave propagation speeds (mean ± STD) from 6.55 ± 0.09 m/s (10 mmHg) to 9.82 ± 0.19 m/s (40 mmHg), leading to an estimate of Young’s modulus, which increased from 145.23 ± 4.43 kPa to 326.44 ± 13.30 kPa. Our implementation of an artificial eye model highlighted that the impact of IOP on Young’s modulus (ΔE = 165.59 kPa, IOP: 10–40 mmHg) was more significant than the effect of stretching of the silicone cornea (ΔE = 15.79 kPa, relative elongation: 0.98–6.49%). Our study sheds light on the potential advantages of using an artificial eye model to represent the response of the human cornea during OCE measurement and provides valuable insights into the impact of IOP on wave-based OCE measurement for future in vivo corneal biomechanics studies.

## 1. Introduction

Corneal biomechanical properties (e.g., stiffness, elasticity, and viscosity) are crucial to maintain the structural stability and visual function of the human eye. Corneal biomechanics—which are affected by normal physiological function, aging, and ocular diseases such as keratoconus [1], glaucoma [2,3], and myopia [4,5,6]—can also be altered by clinical treatments such as refractive surgery [7,8,9,10] and corneal collagen cross-linking [11]. Accordingly, the assessment of corneal biomechanics in a clinical setting would be beneficial in identifying degenerative corneal conditions [12,13], screening refractive surgery candidates [14,15], and evaluating treatment outcomes [16]. However, it remains a long-standing challenge and an active area of research to measure corneal biomechanical properties in vivo [17].

Optical coherence elastography (OCE) is a novel elastic imaging technique that quantifies soft tissue biomechanics using a high-resolution optical coherence tomography (OCT) system to detect the tissue response (e.g., displacements or mechanical waves) under a loading force [18]. Due to the successful implementation of OCT systems in ophthalmology, OCE has been recognized as having great potential in the clinical evaluation of corneal biomechanics, and its development has accelerated over the past decades [19]. Various types of tissue stimulation methods, including mechanical contact [20,21], sound waves [22], pulsed laser [23], air puff/pulse [24,25], and heartbeat stimulation methods [26], have been developed for OCE applications. Among them, the microliter-volume air-pulse simulation method [24] is recognized as a safe and comfortable method for in vivo ocular tissue stimulation due to its non-contact, transient (e.g., as short as ~1 ms), low-pressure (e.g., <60 Pa), and highly localized (e.g., 150 μm stimulation diameter) features. Such stimulation approaches generate micrometer to sub-micrometer tissue displacements, which require a high-resolution OCT system to identify the tissue dynamic response. Structural OCT/OCE uses the amplitude of the complex signals to offer micrometer-scale axial and lateral resolutions. Phase-sensitive OCT/OCE employs the phase signals to further enhance the dynamic elastography detection sensitivity to a sub-nanometer scale [27], which allows for the detection of minute-magnitude dynamics in human corneas in vivo [28,29,30]. The primary indicator of the tissue’s mechanical property is Young’s modulus, a representation of elasticity expressed as the slope between the force (stress) and the resulting fractional deformation (strain). The prevalent OCE technique, similar to ultrasonic elastography, works by inducing mechanical waves in the tissue, tracking the wave propagation, and then estimating Young’s modulus of the tissue based on the wave velocity [19,31,32]. In general, mechanical waves propagate faster in stiffer materials and slower in softer materials. Reliable measurement of the stimulation-induced mechanical waves for the in vivo cornea is the top priority for corneal biomechanical property reconstruction using an OCE system [33], but it remains a challenging task [19].

To date, only a few recent pioneering studies have successfully measured the mechanical wave propagations in human corneas in vivo [28,29,30]. Lan et al. [29] utilized a combination of high-resolution common-path OCT imaging and microliter air-pulse stimulation at a pressure of 13 Pa to induce a submicron displacement amplitude on the corneal surface. They observed and measured the propagation of surface waves in the spatiotemporal domain of 18 eyes from nine healthy individuals (three females and six males) with an average age of 27 ± 5 years and an IOP ranging from 9.3 to 23.2 mmHg. The group velocity of the surface waves ranged from 2.4 to 4.2 m/s, with a mean of 3.5 m/s and a 95% confidence interval of 3.2–3.8 m/s. The results showed a correlation between group velocity and central corneal thickness (r = 0.64, *p* < 0.001) and IOP (r = 0.52, *p* = 0.02) [29]. Ramier et al. [30] evaluated the shear modulus of human corneas by utilizing an OCE system equipped with a vibrational contact probe (diameter: 2 mm) driven by a pair of acoustic transducers (20 mN, frequency: 2–16 kHz). In their study, they measured the Rayleigh-wave speed in 12 healthy individuals (age: 25–67 years, seven males and five females, intraocular pressure (IOP): 13–18 mmHg) to be 7.86 ± 0.75 m/s. However, they did not identify any correlation between the wave speed and IOP or central corneal thickness [30].

These two pioneering OCE studies showed discrepancies in the in vivo corneal mechanical wave measurements, particularly in the relation between mechanical wave speeds and IOPs [29,30]. The discrepancies could have originated from the complex features of corneal biomechanics, as well as the challenges of performing in vivo OCE measurements [17]. The cornea is well known for its nonlinear and anisotropic biomechanical properties, and different stimulation features such as force amplitude and strain-rate can result in variations in shear wave speeds (i.e., 2.4–4.2 m/s [29] and 7.86 ± 0.75 m/s [30]) and different estimations of Young’s moduli (i.e., ~40 kPa [29] versus ~200 kPa [30]). Actually, the estimation of human corneal Young’s modulus can vary from approximately kPa to several tens of MPa, owing to the nonlinear stress–strain behavior of the cornea and the varying techniques and conditions under which the cornea is measured (ex vivo versus in vivo, dehydration states, amplitude or rate differences of the applied force, etc.) [17]. Hence, during corneal biomechanics assessments, it is essential to compare the measurement conditions, in addition to shear wave speeds or Young’s moduli. The controversial correlation studies between the mechanical wave speeds and IOP may have resulted from the challenges of in vivo corneal wave propagation measurement. The primary artifact of in vivo corneal OCE measurements is eye motion [34], which can be greatly influenced by OCE stimulation as well as physiological ocular movements induced by respiration and heartbeats. These factors can lead to significant measurement variability for corneal shear wave propagation speeds. The IOP measurement using the applanation method is also widely acknowledged to be influenced by central corneal thickness (CCT) and corneal biomechanics [35], and the fluctuating IOP values throughout cardiac cycles may also affect the testing conditions or alter the corneal biomechanical properties, hence causing measurement discrepancies over time. In addition, the correlation study between shear wave velocity and IOP was performed among different eyes; but each eye has its own features, including different elasticity, viscosity, and geometry (e.g., CCT). Previous in vivo OCE studies [28,29,30] have not controlled for individual differences in corneal elasticity, viscosity, and geometry (e.g., central corneal thickness), all of which may affect the in vivo OCE measurement results. Therefore, it is possible that these factors contributed to the discrepancies observed among different eyes in the correlation study between shear wave velocity and IOP [29,30].

To investigate the relationship between IOP and mechanical wave propagation under a better-controlled condition, we built a simplified artificial eye model and utilized a microliter air-pulse OCE system to measure the surface wave propagations in the radial directions of the artificial eye’s silicone cornea. The silicone material exhibits a more linear stress–strain relationship in comparison to the human cornea. During OCE measurement, the IOP was increased from 10 mmHg to 40 mmHg by altering the amount of water in the eye model and was monitored precisely by a pressure sensor. The effects of stretching on the elasticity and curvature of silicone corneas were taken into account when calibrating the OCE measurement results. The change in Young’s modulus caused by the stretching of the silicone cornea was evaluated independently by mechanical testing of the silicone material, while the change in corneal curvature at each intraocular pressure was acquired by OCT imaging. We aimed to conduct a more robust comparative analysis of the relationship between surface wave propagation speed and the estimated Young’s modulus with changes in IOP levels.

## 2. Materials and Methods

### 2.1. Artificial Eye Model

Figure 1 illustrates the process to produce the silicone cornea and the whole artificial eye model. In the production of the silicone cornea for an artificial eye model, two types of self-defoaming silica gels were used (A silicone and B silicone from Sanjing Xinde Technology Inc., Beijing, China). These two materials were mixed in equal proportions in a Petri dish (Figure 1a) and settled for 30 min until all air bubbles had dissipated. The resulting silicone mixture was poured into a corneal mold that had been previously coated with a petroleum jelly layer to prevent adherence (Figure 1b). A cap was placed on top of the mold to form the silicone into a desired shape. The mold was then placed in a refrigerator for 1 day to ensure complete solidification of the silicone. The silicone cornea was then removed from the mold, as depicted in Figure 1c. The silicone cornea was composed of a central protrusion portion with a 12 mm diameter to replicate the cornea and a surrounding flat edge (diameter: 12 to 20 mm; thickness: 1.5 mm) that was utilized for attachment to the rest of the eye model. The anterior and posterior radii of the central portion were molded to 7.79 mm and 6.38 mm, respectively, with a central corneal thickness (CCT) of 0.55 mm. A variation in temperature or water pressure within the eye model could result in the variation in these parameters. The remaining parts of the artificial eye model were 3D printed and consisted of a cover (2.3 mm thick; inner diameter: 12.4 mm; outer diameter: 36.3 mm) and a chamber (height: 20 mm; inner and outer diameters: 20.0 mm and 36.3 mm, respectively) that enclosed the silicone cornea’s flat edge (see Figure 1d). The chamber had two channels, which can further link to a water tube and a pressure sensor for controlling IOP (also see Figure 2b).

### 2.2. OCE System Set-Up

A detailed description of this OCE system has been provided in our previous work [36]. In summary, this OCE system was built by combining a microliter air-pulse mechanical stimulation system and a 1290 nm linear-wavenumber (k) spectral domain OCT platform [37] (Figure 2a). Microliter air-pulses (99.99% nitrogen) were delivered to the silicone corneal apex of the artificial eye model via a microbore cannula controlled by a high-speed solenoid valve. The air pulse had a low pressure (~200 Pa), a short duration time (~3 ms), and a small stimulation diameter (150 μm) on the sample surface. The light source in the OCT system was a superluminescent diode (SLD, IPSDS1307C-1311, Inphenix Inc., Livermore, CA, USA) with a 3 dB bandwidth of 1290 ± 40 nm. The light was split 50:50 to the reference and sample arms. In the sample arm, light was collimated to a 4 mm parallel beam, scanned by 2-dimensional galvo mirrors, and then focused by a 54 mm telecentric scan objective (LSM54-1310, Thorlabs Inc., Newton, NJ, USA) with a wide scan field (18.8 × 18.8 mm^2^). The maximum output power was ~1.8 mW measured at the sample surface. The returned light from both arms was interfered and recorded by a linear-wavenumber (k) spectrometer (PSLKS1300–001-GL, Pharostek, Rochester, MN, USA). The combination of the grating and prism (Figure 2a) in the linear-k spectrometer disperses the spectrum linearly in the k domain to improve the signal-to-noise ratio (SNR) in deeper tissue, and to reduce the computing time required to convert the spectrum from the wavelength domain to the wavenumber domain as in a conventional grating-based spectrometer. The k-domain interference signals were captured by an InGaAs line scan camera with 2048 pixels and a line rate of 76 kHz (GL2048L-10A-ENC-STD-210, Sensors Unlimited, Inc., Princeton, NJ, USA). The acquired signals were sent to a computer where they were processed via the Fourier transform to acquire the depth profiles (A-scans) using code written in the LabVIEW language. The OCT imaging system had a 6.94 mm imaging depth, a 99.3 dB maximum sensitivity, and a maximum sensitivity fall-off of −28.6 dB, with a −6 dB fall-off range of ∼0–3 mm. The axial resolution was ~15 µm (all calibrated in air).

Figure 2b demonstrates the artificial eye model during OCE measurement. The eye model was connected to two channels, one linked to a water tube and a reservoir to regulate water volume in the chamber, and the other connected to a pressure sensor (WNK81MA, Hefei WNK Smart Technology Co.,Ltd., Hefei, Anhui, China) for monitoring the chamber pressure (i.e., equivalent IOP). To facilitate OCE mechanical wave measurement, the pressure was controlled in the range of 10 mmHg to 40 mmHg.

### 2.3. M-B Mode Radial Scan Patten

The OCT system was synchronized to the stimulation to observe and quantify the induced dynamic response in the radial directions surrounding the stimulation point (sample apex). Figure 3a shows the OCE-measuring geometry between the stimulation and measurement points. The air pulse was delivered at the silicone corneal apex, and the OCE measurement was performed in each radial direction from 54° to 324°. The angle between adjacent radial directions was 18°, and each direction had 17 measurement points to cover a distance of 0.93–2.05 mm. The measurement was performed using an M-B mode scan protocol to measure the surface displacement as a function of time and to track elastic wave propagation through the tissue surface [32]. The air pulse was stimulated every 0.5 s, and the OCE measurement was synchronized to the stimulation and measured as 11.4 ms at each point to detect the temporal displacement profile of that point. The surface wave propagation speed can be estimated between the time delay of certain distances.

In the complex OCT interferogram signal (${\mathrm{OCT}}_{\mathrm{complex}}$), the phase signal ϕ can be determined by the real and imaginary parts of the OCT signals as follows:(1)$$\emptyset =arctan\left[\frac{Im\left({\mathrm{OCT}}_{\mathrm{complex}}\right)}{Re\left({\mathrm{OCT}}_{\mathrm{complex}}\right)}\right]$$

The temporal profile of phase variation ϕ(t) can usually be resolved and unwrapped by tracing one point in a time sequence (reference to time $t_{0}$) among the successive A-scan signals. Surface displacements y(t) can be converted from the phase signals as follows [38]:(2)$$y\left(t\right)=\frac{\lambda_{0}}{4\pi n}\phi \left(t\right),\;$$
where $\lambda_{0}$ is the center wavelength and n is the refractive index (n=1 in air). In the absence of applied forces, the phase stability (standard deviation) was measured using a mirror that was 4.3 ± 1.4 milliradians over 60 ms (10 repetitive measurements) in the common-path OCT setup, corresponding to displacements of 0.44 ± 0.14 nm.

In OCE measurement, the mechanical wave propagation velocity is directly correlated with the tissue’s mechanical property. In general, the wave propagates faster in a stiffer material and slower in a softer material. Young’s modulus (E) is a measure of elasticity and is represented as the ratio of stress to strain. A larger value of Young’s modulus is associated with greater tissue stiffness and thus faster mechanical wave traveling speed during OCE measurement. In a surface wave equation, the relation between the group velocity (c) and Young’s modulus can be expressed as follows [39]:(3)$$E=\frac{2\rho {\left(1+v\right)}^{3}}{{\left(0.87+1.12v\right)}^{2}}c^{2},\;$$
where ρ is the density and ν is the Poisson’s ratio. Generally, v can be assumed as 0.49. The group velocity of the elastic surface wave was estimated as follows:(4)$$c=\frac{1}{n}\sum_{i=1}^{n}\left(\frac{S_{i}}{t_{i}}\right),\;$$
where $S_{i}$ and $t_{i}$ represent the traveling distance and time delay for the surface wave to travel from the stimulation point to the measurement point i (i=1, …, n), respectively. When the stimulation was on the corneal apex, the traveling distance $S_{i}$ along the corneal surface (arc distance) is
(5)$$S_{i}=R\cdot \arctan\left(\frac{r_{i}}{R}\right),\;$$
where $r_{i}$ is the OCT scan (or horizontal) distance and R is the radius of curvature for the anterior cornea (see Figure 3b). Equation (4) can typically be accomplished using linear curve fitting [29].

## 3. Results

### 3.1. Geometry of the Silicone Cornea in Association with IOP Values

As the pressure within the artificial eye model increases, the silicone cornea is subject to stretching and protrusion, resulting in changes to its radius of curvature and thickness, which can affect the OCE measurement results. To account for this, the structure of the silicone cornea was calibrated at different IOPs, ranging from 0 to 40 mmHg, as shown in Figure 4. A single OCT B-scan was conducted to characterize the geometry variation caused by IOP, capturing a radial cross-section centered at the silicone cornea’s apex. The refractive indexes of the silicone and water were measured as 1.496 and 1.371, respectively, at the wavelength of 1290 nm. Figure 4a–e display cross-sectional OCT images of the silicone cornea at different IOPs, with Figure 4a showing the relaxation condition (0 mmHg) and Figure 4b–e showing the pre-stress condition loaded by water pressure (10–40 mmHg). The silicone corneal diameter (SCD) was 11.76 mm, and the anterior surface depth (ASD), radius of curvature (R), central corneal thickness (CCT), and anterior surface length (ASL) were measured as the IOP increased from 0 mmHg to 40 mmHg (Figure 4f–i). In Figure 4f, the ASD values were measured between the corneal apex and boundary and observed to increase from 2.73 to 3.30 mm (linear fitting equation: y = 0.0142x + 2.726, R^2^ = 0.999) as the IOP increased. The R of the equivalent spherical anterior surface was calculated using $R=\left(4ASD^{2}+SCD^{2}\right)/8ASD$ and was observed to decrease from 7.70 mm to 6.89 mm (linear fitting equation: y = −0.0202x + 7.678, R^2^ = 0.996, Figure 4g). The CCT decreased from 0.545 mm to 0.477 mm (linear fitting equation: y = −1.772x + 551.6, R^2^ = 0.968, Figure 4h) with increasing IOP. Furthermore, the ASL was observed to elongate from 14.59 mm to 15.64 mm (linear fitting equation: y = 0.0238x + 14.53, R^2^ = 0.983, Figure 4i). The calibrated ASL values can be used to estimate changes in the cornea’s mechanical properties (i.e., Young’s modulus) through mechanical testing (see Section 3.2).

As R varied depending on the IOP values, the arc distance along the anterior surface of the silicone cornea between the stimulation point and each measurement point (as illustrated in Figure 3b) also changed. We used the arc distance as the wave propagation distance for the surface wave speed calculation (Equations (4) and (5)). The OCT line scan distance in each radial direction was 0.933–2.053 mm, and the measured arc distances in each radial direction were 0.960–2.519 mm (10 mmHg), 1.018–2.671 mm (20 mmHg), 1.067–2.801 mm (30 mmHg), and 1.130–2.964 mm (40 mmHg), as shown in Figure 5.

### 3.2. Effect of Stretching on Young’s Modulus of Silicone Material

When the water pressure inside the artificial eye model (equivalent to the IOP) increased, the silicone cornea protruded forward, stretching the silicone cornea along radial directions, which caused a change in the elasticity of the silicone material (Figure 6a). To emulate the stretching condition of the silicone cornea caused by increased IOP, we conducted mechanical testing on a silicone block to measure the change in Young’s modulus when the material was stretched, as shown in Figure 6b–f. The measurement was repeated five times. A cylindrical silicone block with a natural diameter of $D_{0}\;$= 44.00 mm and thickness of $H_{0}\;$= 10.58 mm was fixed with eight posts and stretched radially by two positioning discs, as illustrated in Figure 6b–d. The outer diameter of the positioning discs determined the stretched value (ΔD), while the central area of the silicone material was placed between two plates on the top and bottom for compression testing. As the diameter of the cylindrical silicone block was stretched from 44.00 mm ($D_{0}$) to 56.00 mm (increase step: 2 mm, maximum ΔD = 12 mm), the center thickness H was reduced from 10.85 ± 0.05 mm (H_0_) to 9.04 ± 0.06 mm (linear fitting: y=−0.14x+17.10, R^2^ = 0.995, Figure 6e). The measured Young’s modulus of the stretched silicone block was increased from 1001.10 ± 14.64 kPa ($E_{0}$) to 1580.31 ± 50.58 kPa (maximum ΔE = 579.2 kPa, maximum $\Delta E/E_{0}$ = 57.86%). As shown in Figure 6f, the correlation between the relative change in Young’s modulus ($\Delta E/E_{0}$) and the relative diameter elongation value ($\Delta D/D_{0}$: 0–27.27%) was as follows:(6)$$\frac{\Delta E}{E_{0}}=1.973\frac{\Delta D}{D_{0}},\;$$
where the fitting R^2^ = 0.981. Equation (6) can be used to predict the trend of Young’s modulus change in response to the stretching effect and can be used to calibrate the OCE measurement results on Young’s modulus estimation and the correlation study between Young’s modulus and IOP.

### 3.3. Surface Wave Characterization in the Silicone Cornea

The speed of surface waves was determined by measuring the distance and time delay between the stimulation and measurement points (Equations (4) and (5)). Figure 7 provides an example of this calculation for the silicone cornea at 10 mmHg IOP in the 180° direction (as shown in Figure 3a). The measurement was repeated four times. The surface displacement profiles in the temporal and spatiotemporal domains are depicted in Figure 7a,b. The stimulation was applied at the cornea apex, and measurements were taken at 17 sampling points along a scanning distance of 0.933 mm–2.053 mm, covering an arc distance of 0.960–2.519 mm along the surface of the silicone cornea. The induced surface dynamics exhibited three periods [27]: (1) a baseline period before stimulation; (2) a stimulation-force-driven primary deformation period where displacement increased from baseline to the maximum negative displacement and then recovered; and (3) a vibration period, which underwent a decay oscillation and returned gradually to its baseline position. The maximum negative displacement amplitude ($A_{0}$) of each measurement location within the analysis window (4.0–6.8 ms) was assessed to characterize the mechanical wave propagation. Figure 7c shows the decay of $A_{0}$ during the wave propagation process. In this study, the amplitude decreased from −3.50 ± 0.05 µm to −1.38 ± 0.02 µm as the measurement arc distance increased from 0.960–2.519 mm. An exponential function was used to describe the distance-dependent decay of the primary deformation. The curve was fitted as $y=A_{0}e^{B\left(x-0.96\right)}$, where $A_{0}$ is the amplitude, B is the decay coefficient, and the x and *y*-axis scales are in millimeters and micrometers, respectively. The obtained values were $A_{0}=-3.56\;\mu m,\;B=-0.57\;{\mathrm{mm}}^{-1},\;\mathrm{and}\;R^{2}=0.978.$ Figure 7d shows a linear fitting between the time delay and measurement position, and the slope of this curve represents the wave propagation speed along the 180° direction of the silicone cornea at 10 mmHg IOP. The observed wave propagation speed was 6.48 m/s (95% CI: 6.47–6.49, R^2^ = 0.98).

Figure 8 illustrates the propagation of surface waves in the radial directions (54° to 324°) of the silicone cornea at an IOP of 10 mmHg. Figure 8a shows the propagation and attenuation process of the surface wave at varying measurement points on the silicone corneal surface (top view) during different OCE measurement times (4–9 ms). Because the silicone cornea was fabricated uniformly in its radial directions, the propagation waves also presented a uniform geometry surrounding the stimulation point (apex of the silicone cornea). The propagation feature of surface waves is fully illustrated in Appendix A, which can be found in the Appendix A. Figure 8b illustrates the spatiotemporal relation of the surface wave in five propagation directions (90°, 126°, 180°, 234°, and 270°), with velocities ranging from 6.48 ± 0.15 m/s to 6.67 ± 0.31 m/s. Notably, the surface waves propagate consistently with similar displacement magnitudes and propagation speeds in all radial directions in the isotropic samples, as demonstrated in this example.

Figure 9 illustrates the characteristics of surface wave propagation and the estimated Young’s modulus in the radial directions (54° to 324°) of the silicone cornea at IOPs ranging from 10 mmHg to 40 mmHg. The experiment was repeated 4 times. In Figure 9a, the wave propagation speeds (mean ± fitting confident intervals) are presented in different directions and at various IOPs. The waves propagated uniformly in each radial direction with speeds (mean ± STD) of 6.55 ± 0.09 m/s (10 mmHg), 7.06 ± 0.10 m/s (20 mmHg), 7.72 ± 0.07 m/s (30 mmHg), and 9.82 ± 0.19 (40 mmHg). Figure 9b shows the primary magnitude ($A_{0}$) of the displacements. An offset between the measurement center and stimulation center was observed. The displacements distributed evenly surrounding the stimulation center, with measurements of 3.13 ± 0.27 µm (10 mmHg), 2.68 ± 0.22 µm (20 mmHg), 2.29 ± 0.15 µm (30 mmHg), and 2.13 ± 0.15 µm (40 mmHg). Figure 9c shows the correlation between the surface wave speeds and the IOPs (exponential fitting: $y=0.24e^{0.09\left(x-10\right)}+6.37$, R^2^ = 0.997). Figure 9d shows the measured and calibrated Young’s modulus (in kPa) in association with the IOPs (in mmHg). The solid blue line displays Young’s moduli estimated using the surface wave speeds, which were 145.23 ± 4.43 kPa (10 mmHg), 168.73 ± 4.78 kPa (20 mmHg), 201.75 ± 3.66 kPa (30 mmHg), and 326.44 ± 13.30 kPa (40 mmHg). The correlation between the wave-based Young’s moduli and IOP was fitted as an exponential curve, $y=19.45e^{0.1072\left(x-10\right)}+141.3$, R^2^ = 0.992. The dashed green line represents the stretching effect of the silicone material on Young’s modulus change using Equation (6). Predicted linear increments in stretching-induced Young’s moduli ranged from 145.23 kPa (10 mmHg) to 161.02 kPa (40 mmHg), as the relative anterior surface length varied from 0.98% (10 mmHg) to 6.49% (40 mmHg). Furthermore, the orange line in Figure 9d depicts Young’s modulus relative to the IOP change after the stretching effect was removed, which was 145.23 ± 4.43 kPa (10 mmHg), 164.17 ± 4.78 kPa (20 mmHg), 192.23 ± 3.66 kPa (30 mmHg), and 310.82 ± 13.30 kPa (40 mmHg). The correlation between the calibrated Young’s moduli and IOP was $y=13.71e^{0.119\left(x-10\right)}+143.7$, R^2^ = 0.998.

Based on the findings, there was a relationship between mechanical wave speeds, the silicone cornea’s mechanical properties, and IOP. In this artificial eye model study, the stretching effect had a minimal impact on Young’s modulus (ΔE = 15.79 kPa, from 10 to 40 mmHg), while the IOP significantly affected the mechanical wave propagation speed and the resulting estimations of Young’s modulus (ΔE = 165.59 kPa, from 10 to 40 mmHg). Consequently, when applying the wave-based OCE to clinical settings, it is critical to address the impact of IOP on the measurement results for a more accurate assessment of corneal biomechanics.

## 4. Discussion and Conclusions

The relationship between mechanical wave speeds and IOP levels remains a topic of debate [29,30] due to the difficulties in conducting in vivo corneal OCE measurements. To gain a better understanding of this, we constructed an artificial eye model and utilized a microliter air-pulse OCE system to measure the surface wave propagations in the radial directions of the silicone cornea. Employing an artificial eye model offers several benefits, including a simplified and controlled measurement condition with adjustable and monitorable IOP values, as well as the removal of the physiological ocular motions typically induced by respiration and heartbeats. Additionally, using the simplified eye model retains the primary geometry and boundary conditions while eliminating the complex mechanical properties present in the corneal tissue, which can be viscous, anisotropic, and highly nonlinear in the stress–strain response.

During OCE measurement, the IOP increased from 10 mmHg to 40 mmHg by altering the amount of water in the eye model and was monitored by a pressure sensor. As the pressure increased, the silicone cornea stretched further and protruded forward, causing a decrease in its radius of curvature and thickness. To account for these changes, we calibrated the OCE surface wave measurement results using two methods. First, we measured the cornea’s geometry parameters using OCT imaging to calibrate surface wave velocity (refer to Figure 4 and Figure 5). Second, we evaluated Young’s modulus change in the silicone material due to its elongation using a mechanical testing method (Equation (6)). Applying these calibration techniques helped to obtain accurate OCE measurement results while accounting for the changes in geometry and elasticity of the silicone cornea due to water pressure.

As illustrated in Figure 9, the surface wave propagated evenly in the radial directions of the silicone cornea, along a scanning distance of 0.933 mm–2.053 mm (arc distance: from 0.960–2.519 mm at 10 mmHg to 1.130–2.964 mm at 40 mmHg). The measured surface wave velocity increased from 6.55 ± 0.09 m/s to 9.82 ± 0.19 m/s as the IOP increased from 10 to 40 mmHg, resulting in an estimate of Young’s modulus, which increased from 145.23 ± 4.43 kPa to 326.44 ± 13.30 kPa. As the elongation of the silicone material changed Young’s modulus linearly (ΔE = 15.79 kPa, relative elongation: 0.98–6.49%), the calibrated Young’s modulus, after accounting for the effect of elongation, still increased greatly as IOP increased (ΔE = 165.59 kPa, IOP: from 10 mmHg to 40 mmHg). As a result, the stretching had a relatively small impact on Young’s modulus of the cornea. The mechanical wave propagation speed of the cornea was more significantly affected by IOP. Thereby, further studies should be performed to better separate the effect of IOP on Young’s modulus estimation in wave-based corneal OCE application.

This study has certain limitations that may impact the estimation accuracy of the correlation between surface wave speeds and IOPs. First, the use of a single-layered silicone cornea may not fully replicate the behavior of the human eye as IOPs increase. The human cornea is a complex, layered structure with varying stiffnesses in different regions and directions. The stroma, which comprises the majority of the cornea’s thickness, largely determines the overall properties of the human cornea. The orientation and depth-arranged pattern of the collagen fibers/lamellae of the stroma result in the anterior portion of the cornea having the most strength, followed by the middle part, while the posterior part is softest. As a result, the human cornea can uniquely adapt to fluctuations in IOP. Specifically, the inner section of the cornea can change its geometry to accommodate these changes, while the outer layer of the cornea can maintain its shape and preserve the quality of vision. However, this is not the case in this simplified artificial eye model, where the entire silicone cornea changes shape and thickness in response to changes in IOP. Although calibration has been performed to account for changes in corneal shape and elasticity due to pre-stress conditions caused by IOP, this calibration may not fully reflect the actual situation. Because the primary aim of this paper is investigating the correlation between shear wave speeds and IOP in the cornea, using a simplified version of the artificial cornea would be advantageous in order to avoid structural complexity and mechanical nonlinearity. This would enable us to focus more on the relation between shear wave speeds and IOP. In future studies, we will develop multi-layered silicone corneas that can more accurately represent the axial distribution of corneal mechanical properties to better simulate the behavior of the eye when the IOP changes. Second, the mechanical calibration methods may only partially represent the actual elasticity changes of the silicone cornea at various IOP levels. In the mechanical testing, we used a silicone block to mimic the mechanical behavior of the silicone membrane as it was stretched. It should be noted that soft materials, such as the cornea and silicone, typically exhibit non-linear elasticity, which means that the estimated Young’s modulus is lower in the low-strain region (e.g., hundreds of kPa in microliter air-pulse OCE measurement, as illustrated in Figure 9), but higher in the high-strain region (~MPa, as demonstrated in Figure 6, in mechanical testing). Consequently, the linear estimation in Equation (6) between the relative change in Young’s modulus (∆E/E) and the elongation rate (∆D/D) could overestimate the trend of Young’s modulus change when the silicone material is stretched in a low value. Nevertheless, it is important to note that even if the increase in Young’s modulus due to stretching is overestimated, its overall impact on Young’s modulus is relatively small. As a result, the effect of IOP on the corneal mechanical wave speed, as well as the estimated Young’s modulus, far outweighs the increase in Young’s modulus due to the stretching of the silicone material.

In a recent publication, Pitre and colleagues [40] observed the discrepancy between corneal elasticity measurements obtained by OCE methods and other methods, i.e., tensile test measurements. They argued that the cornea’s transversely isotropic structural properties account for the observed differences in tissue elasticity measurements and proposed a novel computational model (nearly incompressible transverse isotropy model, NITI) to account for the differences between shear and tensile moduli. Furthermore, they claimed that the model fails above 30 mm Hg due to the nonlinearity of corneal tissue. A modified version of the NITI model also demonstrated that the shear modulus and spring constant increase non-linearly (exponentially) in response to IOP ranging from 5 to 20 mmHg [41]. Nevertheless, our results show similar responses—nonlinearity above 30 mm Hg—using a simplified silicone corneal phantom where structural anisotropy does not exist. This then raises the question of why we also observe a nonlinear change in elasticity as a function of IOP in a homogenous, isotropic material. A possible explanation is that the rise in IOP above 30 mm Hg resulted in plastic deformation for both the cornea and the silicone phantom. We think this is unlikely. It is possible that the observed nonlinear response is a more fundamental property of mechanical wave propagation in thin shells under inflation pressure. The nonlinear increase in elasticity seen as an IOP inflation forcing a rise above 30 mm Hg may be due to the enhanced complexity of mechanical wave propagation modes possible in thin structures under increased strain. Additional studies are needed to fully explain this phenomenon.

The use of phantom eye models with different complexities can further facilitate advancements in analytical methods and finite element models for OCE studies that aim to better understand corneal biomechanics. Research efforts on corneal biomechanics are currently focused on two fronts: identifying a simple and relevant biomechanical metric for clinical application, and developing comprehensive analytical or finite element models that incorporate corneal anatomy, IOP, biomechanical properties, pathological changes, and clinical innovations, such as surgical procedures [17]. Notably, a wide range of structural complexity exists in finite element models of the cornea, ranging from a single-layer structure [42] to complex models that account for the fiber organization in stroma [43]. Analytical methods typically employ linear approximations, simplified geometry, and boundary conditions for corneal mechanical response analysis. For example, in wave-based elastography, although the measured group or phase velocities may be the same, the shear/Young’s moduli would differ depending on whether the cornea is considered semi-infinite by the shear or surface acoustic model; a thin-plate, isotropic, and viscoelastic tissue by the Lamb wave model; or a thin-plate, transverse isotropic, and elastic tissue by the modified Lamb wave model (see a recent review paper for more details [19]). Despite the wide range of complexities involved in analytical and finite element methods, the OCE-measured samples are limited to either very basic elastic phantoms (such as silicone or agar) or the cornea itself, ex vivo or in vivo, which possesses complex structural and biomechanical properties that have not been fully comprehended by vision scientists, optical engineers, and eye doctors yet. Therefore, the use of eye models featuring varying levels of sophistication in artificial cornea architectures can be useful in matching the complexity of the corresponding analytical or finite element methods, facilitating the verification and modification of these methods, leading to a better understanding of corneal biomechanics through the use of the OCE technique.

Our study sheds light on the potential of using artificial eye models in OCE research for corneal biomechanics, allowing for greater control in studying the relationship between mechanical wave propagation and IOP changes. Through the implementation of the simplified eye model, we discovered that the impact of IOP on corneal mechanical wave propagation (and Young’s modulus estimation) is more significant than the effect of stretching of the silicone sample when using the wave-based OCE measurement. Therefore, in translating the wave-based OCE to clinical applications, it is crucial to pay close attention to how to remove the influence of IOP to evaluate its Young’s modulus. To better understand the roles of corneal structure and biomechanics on the mechanical behaviors, we intend to start with a simple model and gradually develop more sophisticated eye models that better represent additional structural features and can model additional complexities of corneal biomechanics. During this process, we also hope to gradually improve the analytical methods and the finite element models that better characterize corneal biomechanical properties from the mechanical behaviors. With the continuous advancement of OCE imaging technology and analytical methodologies, we hope to further improve the understanding of corneal biomechanics, facilitating enhanced diagnosis and treatment of ocular diseases.

## Figures and Tables

**Figure 1 bioengineering-10-00754-f001:**
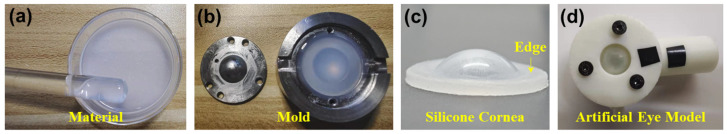
Demonstration of the production process for the artificial eye model. (**a**–**c**) Production of the silicone cornea using the mold. (**d**) The silicone cornea is sandwiched by a cover and a camber to form the artificial eye model. The chamber has two channels, which can further link to a water tube and a pressure sensor (also see Figure 2b).

**Figure 2 bioengineering-10-00754-f002:**
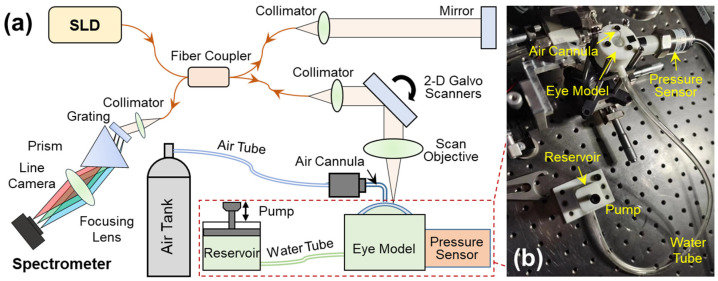
Set-up for the optical coherence elastography (OCE) system and the artificial eye model. (**a**) A microliter air pulse is used to stimulate the silicone cornea perpendicularly, and a spectrum domain OCT system is used to track the micron-scale surface waves. The SLD is a superluminescent laser diode with a waveband of 1290 ± 40 nm; a linear-wavenumber (k) spectrometer was used to disperse the interference spectrum in the k-domain prior to the Fourier transform for OCT processing. (**b**) The artificial eye model comprises a silicone cornea and water chamber connected to a chamber connected to two channels, one linked to a water tube and a reservoir to regulate water volume in the chamber, and the other connected to a pressure sensor for monitoring the equivalent IOP inside the chamber.

**Figure 3 bioengineering-10-00754-f003:**
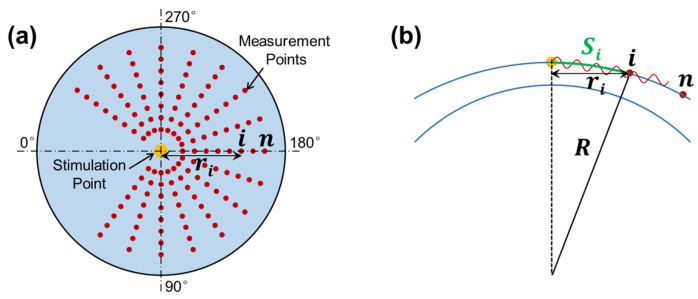
Radial scan pattern for the measurement of surface wave propagation. (**a**) Stimulation and measurement geometry. Air pulse stimulation was performed at the apex of the silicone cornea, and measurement was performed at each radial direction from 45° to 315°. (**b**) Surface wave propagation distance (arc distance: $S_{i}$) in response to corneal curvature (R) and radial distance ($r_{i}$ ).

**Figure 4 bioengineering-10-00754-f004:**
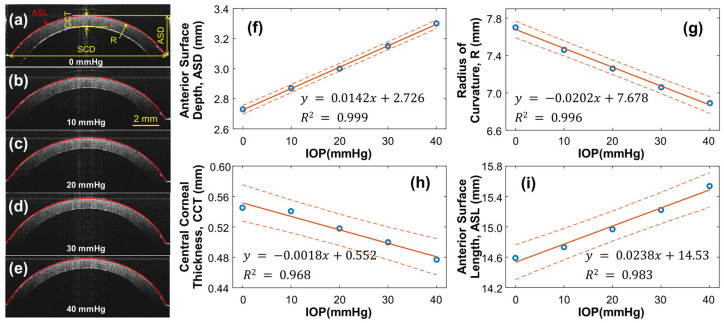
Silicone cornea geometry calibration under 0–40 mmHg IOPs. (**a**–**e**) Silicon cornea OCT imaging at various IOP levels. The anterior surface is segmented for cornea geometry calculation. SCD: Silicone corneal diameter. (**f**–**i**) IOP-dependent anterior surface depth (ASD), radius of curvature (R), central corneal thickness (CCT), and anterior surface length (ASL). Blue circles represent the measurement data, orange solid lines represent the linear fitting results, and the orange dot lines represent the confidence intervals.

**Figure 5 bioengineering-10-00754-f005:**
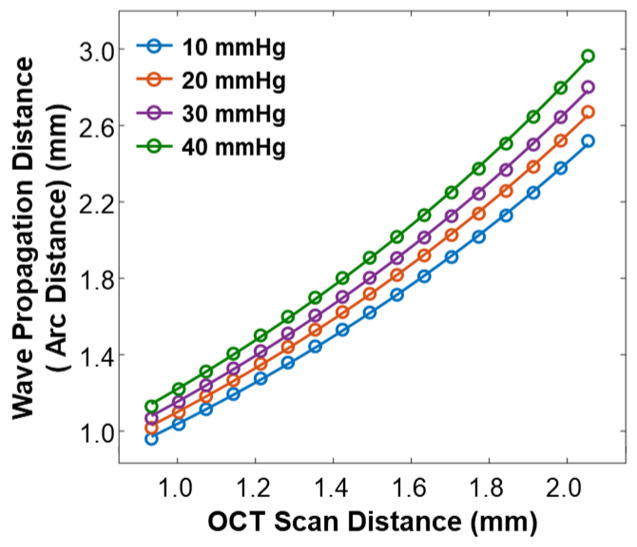
Calibration between the OCT line scan distance and the arc distance of the silicone cornea at various IOP values from 10 mmHg to 40 mmHg.

**Figure 6 bioengineering-10-00754-f006:**
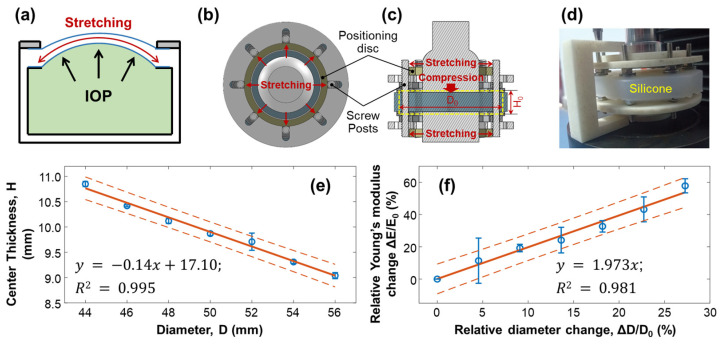
Mechanical testing of silicone material to replicate the change in Young’s modulus due to stretching. The measurements were repeated 5 times. (**a**) The cornea of an artificial eye model protrudes forward under water pressure, leading to pre-stress conditions along the radial directions of the silicone cornea. The increase in Young’s modulus of the silicone material was estimated in panels (**b**–**f**). Panels (**b**–**d**) show the top view, sectional view, and photo imaging of the testing conditions, where a cylindrical silicone block (diameter: D; thickness: H) was fixed with eight posts and stretched radially by two positioning discs positioned on the top and bottom of the testing material. The outer diameters of the positioning discs controlled the stretched value (ΔD), while the central area of the silicone material was placed between a compressor on top and a metal plate below, both of which had a diameter of 30 mm. (**e**) The diameter was stretched from 44 mm ($D_{0}$ ) to 56 mm (maximum ΔD  = 12 mm), reducing the center thickness from 10.85 mm to 9.04 mm. (**f**) The correlation between the relative change in Young’s modulus ($\Delta E/E_{0}$ ) and the relative diameter elongation value ($\Delta D/D_{0}$: 0–27.27%) showed an increase in Young’s modulus (maximum Δ*E*/*E*_0_ = 57.86%, five repeat measurements). Blue circles represent the measurement data (mean ± standard deviation), orange solid lines represent the fitting results, and the orange dot lines represent the confidence intervals.

**Figure 7 bioengineering-10-00754-f007:**
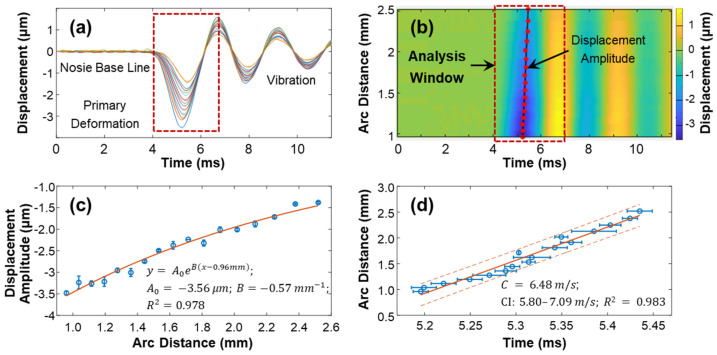
Demonstration of the surface wave speed quantification method for the silicon cornea (10 mmHg) at the 180° direction. (**a**) Surface displacement in the temporal domain. Color series: 17 measurement points at an OCT scan distance of 0.933–2.053 mm, covering an arc distance of 0.960–2.519 mm. (**b**) Spatiotemporal map of the surface wave propagation. The displacement magnitudes were used for magnitude decay fitting in (**c**) and mechanical wave propagation speed fitting in (**d**) by 4 repeated measurements. Blue circles represent the measurement data (mean ± standard deviation), orange solid lines represent the fitting results, and the orange dot lines represent the confidence intervals.

**Figure 8 bioengineering-10-00754-f008:**
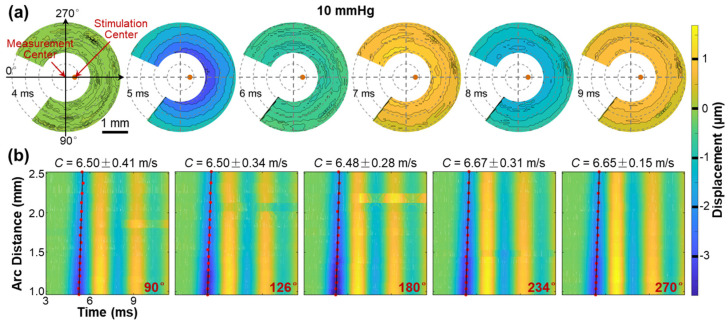
Surface wave propagation in the radial directions (54° to 324°) of the silicone cornea (IOP: 10 mmHg). See Appendix A. (**a**) En face surface wave profiles at different times from 4 ms to 9 ms. (**b**) Spatiotemporal profiles of the surface wave propagation at five selected directions. The wave propagation speeds were calculated by linear fitting for the maximum displacement amplitudes.

**Figure 9 bioengineering-10-00754-f009:**
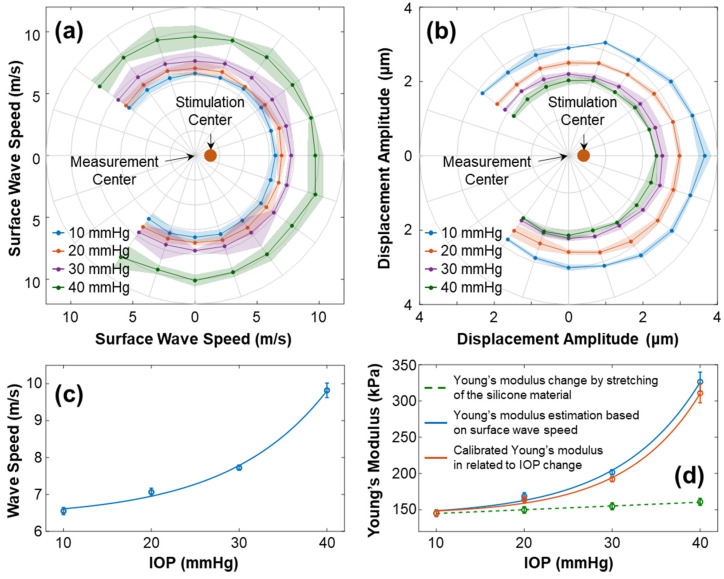
Characterization of the surface wave and Young’s modulus for the silicone cornea at different IOPs from 10 to 40 mmHg. The measurements were repeated 4 times. (**a**,**b**) Directional-dependent wave propagation speeds and displacement amplitudes ($A_{0}$) for 54–324° angles of the silicone cornea at varying IOP levels. (**c**) Mean ± STD of surface wave propagation speeds associated with different IOPs. (**d**) Measured and calibrated Young’s moduli (mean ± STD) in association with IOPs. The dashed green line depicts the trend of Young’s modulus increase caused by material stretching (Equation (6)). The solid blue line shows the estimated Young’s moduli using surface wave speed in (**c**) and Equation (3), while the orange line shows the calibrated Young’s modulus relative to the IOP change.

## Data Availability

The original contributions presented in the study are included in the article/Appendix A, further inquiries can be directed to the corresponding authors.

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
