# Peer review of "Corneal Surface Wave Propagation Associated with Intraocular Pressures: OCT Elastography Assessment in a Simplified Eye Model"

_bioengineering, 2023, doi:10.3390/bioengineering10070754_

Round 1

Reviewer 1 Report

The authors developed an artificial eye model with a silicon cornea. The results are well presented but fail to bring up the advantage of their method. Within the journal's scope, the paper's objective should have been to develop an artificial cornea that is as close as possible to the actual cornea so that such a model can be used to study the effects of various parameters on wave propagation. But the developed cornea is very basic and far from the real one which will not provide any helpful insights. The authors themselves mentioned at many locations that from previous studies, there are discrepancies in the results obtained between ex vivo and in vivo experiments.  How would someone believe that the current model will provide reliable results when the artificial cornea is so different than the real one. Overall I believe that the current manuscript does not add significantly useful information to the field

Reviewer 2 Report

This article presents experimental research on the evaluation of surface wave propagation in artificial corneas under different intraocular pressures using optical coherence elastography (OEM). The authors developed an artificial eye model, referred to as a corneal phantom, by utilizing silica gels that mimic the structure of a real cornea along with water/pressure channels. The OCE system employed in this study was based on their previous work. The measurements of surface waves were robust, and the results were analyzed using appropriate equations and theoretical principles. The research work has its significance in the field of OCE. The manuscript itself was well-prepared and organized. The conclusions drawn by the authors are supported by the experimental results and analysis. Therefore, I support the publication of this article in the Bioengineering. However, there are a few minor errors that need to be addressed before the manuscript can be accepted.

Page 2, line 72: There should be a space after [20].

Page 3, line 143: "Figure 1a" should be corrected to "Figure 2a."

Page 4, line 146: "A... a small diameter (on the sample surface?)" needs clarification or rephrasing.

Page 4, line 155: "Figure 1a" should be corrected to "Figure 2a."

Page 4, lines 174 and 178: "pressor" should be changed to "pressure."

Page 6, line 220: Add "and" after the second comma.

Page 7, line 279: "Figure 5(b-d)" should be corrected to "Figure 6(b-d)."

Page 9, line 324: "During wave propagation" should be changed to "During the wave propagation."

Overall, the language of this manuscript is good. There are some minor errors.

Reviewer 3 Report

1. While optical coherence chromatography is a critical component of the research, it is not mentioned in the abstract.

2. Where is the data about the 3D printer used to create the artificial eye model located in the materials and methods section?

3. The error bars should be included on all data points in the curve of each graph to properly display the degree of uncertainty associated with the results.

4. The section of results does not specify whether the trials were duplicated or triplicated, and the results do not include any standard deviation, which could impact the reliability and reproducibility of the research findings.

1- The article requires revisions to its structure and grammar, as exemplified by issues present in line 36.

Round 2

Reviewer 1 Report

I am still struggling to understand the novelty and impact of the work. As authors claim that the main idea of the work is to understand the relation between the IOP and the shear wave propagation but there are several studies that have already been published with actual cornea (please see references below) both ex vivo and in vivo that show the relation between IOP and shear wave propagation. Unfortunately, for me the results are still not sufficiently novel to recommend publication.

https://doi.org/10.1016/j.jmbbm.2022.105100

Lan G, Aglyamov SR, Larin KV, Twa MD. In Vivo Human Corneal Shear-wave Optical Coherence Elastography. Optom Vis Sci. 2021 Jan 1;98(1):58-63. doi: 10.1097/OPX.0000000000001633. PMID: 33394932; PMCID: PMC7774819.

Author Response

Thank you for your question and your contribution to our research that helped us to refine our work. The supplementary explanations about the novelty and usefulness of using eye models are show as follows.

  • Previous studies have suggested that the anisotropy of the cornea may be a significant factor contributing to the discrepancy between corneal elasticity measurements obtained through OCE and other methods, such as tensile testing. Recently, a nearly-incompressible transverse isotropy (NITI) model and a modified NITI model have been proposed to describe corneal biomechanics using two moduli: shear and tensile moduli. These models have demonstrated that the shear modulus and spring constant of the cornea increase non-linearly in response to intraocular pressure (IOP) ranging from 5 to 20 mmHg, and the models fail above 30 mmHg due to the nonlinearity of the corneal tissue. In this study, we utilized a simplified eye model that retained the main geometry and boundary conditions while removing the complex mechanical properties of the cornea. Our findings showed similar results to those of the NITI papers. This raises the question of why we also observed a nonlinear change in elasticity in a homogeneous, isotropic material. One possible explanation could be that the rise in IOP above 30 mmHg led to plastic deformation for both the cornea and the silicone phantom. However, we believe that this is unlikely. It is possible that the observed nonlinear response is a fundamental property of mechanical wave propagation in thin shells under inflation pressure. The nonlinear increase in elasticity seen as IOP inflation forces rise above 30 mmHg may be due to the enhanced complexity of mechanical wave propagation modes possible in thin structures under increased strain. This part has been described in Lines 501-520.
  • We believe that the use of phantom eye models with different complexities (either a simplified one in this paper, or a more complicated one in our work) can facilitate advancements in analytical methods and finite element models for OCE studies. We have added a new paragraph in the Lines 521-545 to describe this. “The use of phantom eye models with different complexities can further facilitate advancements in analytical methods and finite element models for OCE studies that aim to better understand corneal biomechanics. Research efforts on corneal biomechanics are currently focused on two fronts: identifying a simple and relevant biomechanical metric for clinical application, and developing comprehensive analytical or finite element models that incorporate corneal anatomy, IOP, biomechanical properties, pathological changes, and clinical innovations, such as surgical procedures [17]. Notably, a wide range of structural complexity exists in finite element models of cornea, ranging from a single layer structure [42] to complex models that account for the fiber organization in stroma [43]. Analytical methods typically employ linear approximations, simplified geometry, and boundary conditions for corneal mechanical response analysis. For example, in wave-based elastography, although the measured group or phase velocities may be the same, the shear/Young's moduli would differ depending on whether the cornea is considered semi-infinite by the shear or surface acoustic model, a thin-plate, isotropic, and viscoelastic tissue by the Lamb wave model, or a thin-plate, transverse isotropic, and elastic tissue by the modified Lamb wave model (see a recent review paper for more details [44]). Despite the wide range of complexities involved in analytical and finite element methods, the OCE measured samples are limited to either very basic elastic phantoms (such as silicone or agar) or the cornea itself, ex vivo or in vivo, which possesses complex structural and biomechanical properties that have not fully comprehended by vision scientists, optical engineers, and eye doctors yet. Therefore, the use of eye models featuring varying levels of sophistication in artificial cornea architectures can be useful in matching the complexity of the corresponding analytical or finite element methods, facilitating the verification and modification of these methods, leading to a better under-standing of corneal biomechanics through the use of the OCE technique.”
  • We also made several minor changes, such as refining the references, modification of sentences in Lines 436-439, Line 505, Lines 507-509, and Lines 557-559.